# A Structure—Activity Relationship Study of Bis-Benzamides as Inhibitors of Androgen Receptor—Coactivator Interaction

**DOI:** 10.3390/molecules24152783

**Published:** 2019-07-31

**Authors:** Tae-Kyung Lee, Preethi Ravindranathan, Rajni Sonavane, Ganesh V. Raj, Jung-Mo Ahn

**Affiliations:** 1Department of Chemistry and Biochemistry, University of Texas at Dallas, Richardson, TX 75080, USA; 2Departments of Urology and Pharmacology, University of Texas Southwestern Medical Center at Dallas, Dallas, TX 75390, USA

**Keywords:** α-helix mimetics, bis-benzamide scaffold, protein–protein interaction, prostate cancer, androgen receptor, coactivator PELP1

## Abstract

The interaction between androgen receptor (AR) and coactivator proteins plays a critical role in AR-mediated prostate cancer (PCa) cell growth, thus its inhibition is emerging as a promising strategy for PCa treatment. To develop potent inhibitors of the AR–coactivator interaction, we have designed and synthesized a series of bis-benzamides by modifying functional groups at the N/C-terminus and side chains. A structure–activity relationship study showed that the nitro group at the N-terminus of the bis-benzamide is essential for its biological activity while the C-terminus can have either a methyl ester or a primary carboxamide. Surveying the side chains with various alkyl groups led to the identification of a potent compound **14d** that exhibited antiproliferative activity (IC_50_ value of 16 nM) on PCa cells. In addition, biochemical studies showed that **14d** exerts its anticancer activity by inhibiting the AR–PELP1 interaction and AR transactivation.

## 1. Introduction

Prostate cancer (PCa) is one of the leading causes of cancer death worldwide, accounting for an estimated 1.28 million new cases and 358,000 deaths in 2018 [1]. Initiation and progression of PCa are dependent on the androgen receptor (AR)-mediated signaling pathway triggered by androgens [2]. Upon androgen binding, AR undergoes a conformational change in the ligand-binding domain (LBD) to form a hydrophobic cleft, termed activation function-2 (AF-2), that is recognized by AR coactivator proteins. AR subsequently translocates to the nucleus and activates transcription of the AR related genes leading to cell proliferation [3].

Various strategies have been developed to deprive androgens or block their effects. Surgical or chemical castration suppresses PCa growth by lowering circulating androgen levels. Chemical castration is achieved by blocking testicular and adrenal androgen synthesis with luteinizing hormone-releasing hormone (LH-RH) analogues [4]. Alternatively, antiandrogens have been developed to inhibit AR activities by competitively blocking androgens from binding to AR LBD [3]. Although antiandrogens are initially effective to suppress tumor growth, PCa ultimately turns into an incurable androgen-resistant state [5]. The transition to androgen-resistant state frequently involves AR mutations, overexpression of AR and its splice variants, increased production of intratumoral androgens, upregulation of AR coactivators and so on [5,6].

AR coactivators enhance the transcriptional activity of AR via multiple mechanisms including stabilization/cellular trafficking of AR, chromatin remodeling, and recruitment of general transcription factors [7]. Many coactivators interact with AR through an α-helical LXXLL motif in which L is a leucine and X is any amino acid. The side chains of three leucines at the *i*, *i* + 3, and *i* + 4 positions in the LXXLL motif fit in the hydrophobic pocket at the AF-2 domain of AR [8]. The interaction between the LXXLL motif and the AF-2 domain results in the AR-mediated gene transcription [8], thus the LXXLL motif is a high potential target to suppress PCa growth and overcome drug resistance in PCa [9,10]. As a coactivator, proline-, glutamic acid-, and leucine-rich protein 1 (PELP1) enhances the function of nuclear receptors like AR by coupling them with various signaling factors including transcriptional, chromatin, cytoskeleton, and cell cycle regulators. PELP1 is often found to be overexpressed in several cancers including PCa, and its dysregulation contributes to therapy resistance [11,12].

Short peptide segments encompassing the LXXLL motif can block interactions between the LXXLL motif and the AF-2 domain [13]. However, their therapeutic use is compromised by the intrinsic properties of peptides, such as rapid metabolic degradation, low bioavailability, and poor cell permeability [14]. To overcome these drawbacks, we previously developed oligo-benzamide-based α-helix mimetics that can place its substituents in the same spatial arrangement found in an α-helix, thereby reproducing the structure and function of the helix [15]. Nonpeptidic α-helix mimetics offer advantages of proteolytic stability and cell permeability over natural peptide segments [16]. Previously, we designed a bis-benzamide **D2** with two isobutyl groups based on the canonical LXXLL motif [10]. It disrupts AR–PELP1 interaction in PCa cells, and inhibits transcription and proliferation, suggesting the utility of the bis-benzamide as a therapeutic candidate in PCa treatment [10].

We herein report a structure–activity relationship study of the bis-benzamide and its analogs as potent inhibitors of the AR–coactivator interaction. Cell-based assays identified potent compounds with higher antiproliferative activity compared to **D2** in PCa cells. Further biochemical experiments demonstrated that these compounds were able to disrupt the AR–coactivator interaction and inhibit AR transactivation.

## 2. Results

For the structure–activity relationship study, we focused on three positions in the structure of the bis-benzamide **D2**: the N-terminal nitro group, the C-terminal methyl ester, and two isobutyl substituents at the O-alkylated side chains (Figure 1). To explore the effects of N-terminal substituents, a series of bis-benzamides with different substituents were prepared starting from **D2** that was synthesized by making an amide bond between 3-isobutoxy-4-nitrobenzoyl chloride **1b** and methyl 4-amino-3-isobutoxybenzoate **2** (Scheme 1a) [10]. The nitro group was reduced with tin (II) chloride to make the corresponding amine **4**. Coupling of the aromatic **4** with acyl chlorides produced compounds (**5a** and **5b**) containing N-acylamido groups. Compound **5c** containing a carboxylic acid was obtained by reacting the amine **4** with succinic anhydride. Coupling Boc-Gly to the amine **4** and removing the Boc protecting group from the resulting compound **5d** gave amine-containing compound **5e**. Compound **3** with no substituent at the N-terminus was synthesized from 3-isobutoxybenzoyl chloride **1b** (Scheme 1a).

We next synthesized bis-benzamides with different substituents at the C-terminus. These compounds were prepared from compound **7a** [10] (Scheme 1b). The allyl ester of compound **7a** was removed with Pd(PPh_3_)_4_ affording the carboxylic acid **8** [17]. Treatment of **8** with ammonium chloride [18] and isobutylamine in the presence of PyBOP provided the primary carboxamide **9a** and the N-isobutyl amide **9b**, respectively. Another carboxylic acid-containing compound **9d** was also obtained by introducing a glycine as a spacer at the C-terminus. The carboxylic acid **8** was reacted with glycine allyl ester to yield compound **9c**, of which the allyl group was removed with Pd(PPh_3_)_4_ giving compound **9d**. An amino group (**9f**) was introduced by coupling compound **8** with N-Boc-protected ethylenediamine and removing the Boc group from compound **9e**. Compound **7b** with no substituent at the C-terminus was obtained from 2-isobutoxyaniline **6b** (Scheme 1b). 

These bis-benzamides were examined for their antiproliferative activity on LNCaP cells by MTT assays which quantify cell viability by measuring the activity of mitochondrial enzymes in live cells that reduce MTT (Table 1). Elimination of the N-terminal nitro group (**3**) or its replacement with either an amino (**4**) or N-acylamido group (**5a** or **5b**) led to a significant reduction in the inhibitory activity compared to **D2**. Introducing polar substituents at the N-terminus, such as a carboxylic acid (**5c**) or an aliphatic amine (**5e**) also resulted in complete loss of activity. These data suggest that the nitro group is critical for the biological activity of **D2**.

Among the substituents at the C-terminus, the carboxylic acid **8** was found to be moderately potent (IC_50_ = 90 nM) while the primary carboxamide **9a** (IC_50_ = 57 nM) showed improvement from the carboxylic acid **8**. In fact, it is comparable to **D2** (IC_50_ = 40 nM). However, other C-terminal amide derivatives containing isobutyl (**9b**), carboxylic acid (**9d**), or aliphatic amine (**9f**) did not show any activity (Table 1). The primary carboxamide **9a** appears to be a promising lead since carboxamides tends to have favorable properties, such as superior proteolytic stability [19] and aqueous solubility [20] when compared with methyl esters.

Next, we examined the effect of the side chains of the bis-benzamide **9a** by constructing and evaluating a small library of bis-benzamides. Since the side chains of the LXXLL motifs make a hydrophobic surface to interact with the AF-2 domain of the AR [8], we focused on hydrophobic groups as substitutions. Larger hydrocarbon chains compared to the isobutyl group of the leucine residues of the LXXLL motifs may cause steric clash in the AF-2 domain. Indeed, compounds containing isopentyl or benzyl groups at the side chains were found to be inactive in the MTT assay (data not shown). Therefore, five alkyl groups of identical or a smaller size than the original isobutyl moiety, such as n-propyl, isopropyl, n-butyl, isobutyl, and sec-butyl groups, were selected to generate a 24-member bis-benzamide library, from which a compound containing two isobutyl groups (**9a**) is excluded (Scheme 2).

The library synthesis commenced with the loading of 3-alkoxy-4-nitrobenzoic acids **10** onto Rink amide resin (Scheme 2). The nitro group of **11** was then reduced with tin (II) chloride under acidic conditions. The resulting amines **12** was reacted with 3-alkoxy-4-nitrobenzoic acids **10** using HATU to form a resin-bound bis-benzamides **13**. After cleavage with TFA, 24-membered bis-benzamide library **14** was prepared in high purity (92%–99%) [21]. 

The bis-benzamide library **14** was screened to evaluate their antiproliferative activity at 200 nM by MTT assays (Figure 2). Using a cutoff at 80% inhibition, 13 compounds were selected for follow-up studies. Dose-response experiments were carried out for the selected compounds, and their activities are summarized in Table 2. Among them, **14d** and **14s** were found to be highly potent with the IC_50_ values of 16 and 24 nM, respectively (Figure 3). Additionally, compound **14d** exhibits approximately 6-fold increase in inhibitory activity compared to compound **9a** (IC_50_ = 90 nM) and 2.5-fold increase to the original compound **D2** (IC_50_ = 40 nM). 

Compounds **14f**, **14h**, **14i**, and **14l** showed the highest efficacy resulting in complete inhibition at 200 nM (Figure 2), however their IC_50_ values range from 66 to 81 nM (Table 2). On the other hand, compounds such as **14c**, **14m**, **14o**, and **14r** exhibited the inhibitory activity with the IC_50_ values of 44–57 nM. Compounds **14d**, **14s**, and **14o** demonstrated that the order of substituents at the side chains seems to be important to achieve high inhibitory activity, as observed in total loss of inhibition of PCa proliferation when the order of substituents is reversed (e.g., **14p**, **14w**, and **14v**) (Figure 2). These results suggest that the interaction of the bis-benzamides on the AF-2 domain of AR is specific, and topology of the side chain groups is critical for high affinity. However, isopropyl and n-butyl groups (**14h** and **14l**) are interchangeable without a considerable loss in potency (Table 2).

To determine if **14d** and **14s** can block the AR–coactivator interaction, we carried out co-immunoprecipitation (co-IP) experiments of AR and PELP1 on LNCaP cells (Figure 4a). Treatment of LNCaP cells with **14d** blocked 5α-dihydrotestosterone (DHT)-induced AR–PELP1 complex formation, resulting in complete inhibition at a concentration of 100 nM. Compound **14s** also inhibited the protein complex formation, but with slightly less efficiency.

After AR associates with coactivators, the complex subsequently enhances transcription of AR target genes [3]. Thus, the effects of **14d** and **14s** on AR transactivation were assessed by luciferase activity on LNCaP cells transfected with ARE-luciferase reporter gene (Figure 4b). Compounds **14d** and **14s** significantly attenuated DHT-induced transcriptional activity with their IC_50_ values in the nanomolar range. Expression of well-known AR target genes such as prostate specific antigen (PSA) and transmembrane protease serine 2 (TMPRSS2) was also measured by quantitative reverse transcription PCR (qRT-PCR). Compound **14s** significantly reduced the expression of PSA and TMPRSS2 mRNA (5.2- and 5.0-fold, respectively) in the presence of DHT. Similarly, compound **14d** decreased PSA and TMPRSS2 mRNA levels by 2.7- and 2.3-fold, respectively (Figure 4c). These results indicate that blockage of the AR–coactivator interaction and AR transactivation accounts for the antiproliferative activities of compounds **14d** and **14s**.

We next examined the specificity of compounds **14d** and **14s** for the cell growth inhibition. Compounds **14d** and **14s** have little effect on the transcription of AR target genes in the absence of DHT (Figure 4c). In addition, cell proliferation was unaffected by **14d** and **14s** in the absence of DHT (Appendix A), suggesting compounds **14d** and **14s** are dependent on androgen for their inhibitory activity. The specificity of compounds **14d** and **14s** for the cell growth inhibition was also tested using AR-negative PC3 prostate cancer cells. While **14d** and **14s** were effective in inhibiting AR-positive LNCaP cells (Figure 3), they had no effect on the growth of PC3 cells, confirming that the antiproliferative properties are specific to AR-dependent PCa cells (Appendix A).

To investigate the binding mode of compound **14d** to the AF-2 domain of AR, molecular docking studies were carried out by using AutoDock Vina (version 1.1.2, The Scripps Research Institute, La Jolla, CA, USA) [22]. Docking calculations were performed five times with random seed numbers, and 20 conformers from each docking were collected. The resulting 100 conformers of **14d** were clustered by using clustering of conformer’s script in Maestro (version 9.1, Schrödinger, LLC, New York, NY, USA, 2010) (see Appendix A for the mean binding energy value, the number of conformers, and the lowest energy binding mode of each cluster). The docked conformations were further minimized in the MacroModel suite of Maestro (OPLS-2005 force filed). 

Representative binding modes of **14d** and **D2** are shown in Figure 5. Like **D2** and the LXXLL motifs, compound **14d** interacts with the AF-2 domain by projecting its side chains (e.g., n-propyl and isobutyl groups) into the pockets designated for the leucine residues at the *i* and *i* + 4 positions of the LXXLL motif (Figure 5). However, compounds **14d** and **D2** show subtle differences in the binding mode. Compound **14d** appears to make hydrogen bonds to Lys720, whereas **D2** makes a hydrogen bond to Arg726 (Figure 5b,c). In particular, whereas the methyl ester moiety of **D2** is exposed to solvent (Figure 5c), the primary carboxamide of **14d** is buried inside the cavity near the pocket (Figure 5b). This additional interaction may explain the stronger affinity of **14d** to AR and the increased cell growth inhibition activity of **14d** compared to **D2**. 

## 3. Discussion

The interaction between AR–coactivators presents a viable and orthogonal target for overcoming PCa resistance to current antiandrogens. Several synthetic molecules targeting the interaction have been reported based on scaffolds like pyrimidines [23], dihydroperimidine [24], and diarylhydrazides [25]. However, these molecules showed only weak inhibition even at micromolar levels. Thus, there is a continuing need to develop novel and potent inhibitors.

We previously developed a nonpeptidic oligo-benzamide scaffold mimicking the structure and function of α-helical segments [15]. Designed to mimic the LXXLL motif, a bis-benzamide **D2** effectively inhibited the PCa cell growth by blocking the AR–coactivator interaction and AR-mediated gene transcription. This bis-benzamide scaffold is particularly attractive as a therapeutic candidate because of its favorable properties including metabolic stability, cell permeability, and bioavailability [10]. 

In this study, we performed a structure-activity relationship study of the bis-benzamide to identify structural requirements and to improve anticancer activity. N-terminal functionality had a dramatic effect on antiproliferative activity on LNCaP cells. A nitro group of **D2** was found to be critical for the inhibitory activity, whereas compounds possessing either no substituent (**3**) or an amine (**4**) were inefficient to inhibit the PCa cell growth. N-acylamido derivatives containing methyl (**5a**), n-butyl (**5b**), a carboxylic acid (**5c**) or an aliphatic amine (**5e**) also failed to show any activity (Table 1). 

On the other hand, substitution at the C-terminus is more tolerable than at the N-terminus. A carboxylic acid (**8**) and a primary carboxamide (**9a**) at the C-terminus retained inhibitory activity, whereas compounds possessing no substituent (**7b**) or other N-alkyl amides (**9b**, **9d**, and **9f**) were found to be inactive (Table 1). Carboxamides are often preferred over methyl esters when considering proteolytic stability [19] and aqueous solubility [20]. Furthermore, the carboxamide at the C-terminus allowed us to use solid-phase synthetic techniques for the construction of a small library of bis-benzamides [26].

We next investigated the effects of the side chains of compound **9a** on the inhibitory activity. Since substituents larger than isobutyl groups or aromatic groups showed no significant inhibitory activity (data not shown), five alkyl chains (n-propyl, isopropyl, n-butyl, isobutyl, and sec-butyl groups) were selected as side chains, and a small bis-benzamide library was constructed by combination of different side chain groups. Single-dose screening and follow-up dose-response experiments identified that compounds **14d** and **14s** are the most potent with their IC_50_ values of 16 and 24 nM, respectively, showing significant improvement over **D2** and **9a** (Figure 2 and Table 2). It is noteworthy that the n-propyl group of **14d** at R_1_ position and sec-butyl group of **14s** at R_2_ position have higher activity than isobutyl groups at the corresponding positions.

As shown previously, **D2** exerts its anti-proliferative effect by inhibiting the AR–coactivator interaction and AR transactivation [10]. We examined whether **14d** and **14s** have a mechanism of action similar to that of **D2**. Co-IP experiments showed that **14d** was able to block the AR interaction with PELP1 in the presence of DHT at a dose of 100 nM (Figure 4a). In AR luciferase and qRT-PCR assays, **14d** and **14s** were also effective in inhibiting DHT-induced AR transactivation in the nanomolar range (Figure 4b). Importantly, the compounds were found to depend on androgen and AR for the inhibitory activity as shown by gene expression assays in the absence of DHT (Figure 4c) and cell viability assays using AR-negative PC3 cells (Appendix A).

On the other hand, compound **14s** had similar effects on AR transactivation compared to **14d**, but only weakly inhibited the recruitment of PELP1 to AR (Figure 4a). Since a number of coactivators including steroid receptor coactivators (SRCs) also possess the LXXLL motif for their interactions with AR and enhance the transcriptional activity of AR [8], compound **14s** may block such coactivators in addition to PELP1 to display its inhibitory activity on AR transactivation.

## 4. Materials and Methods 

All chemical reagents and solvents were obtained from commercial sources and used without additional purification. Thin-layer chromatography (TLC) was performed on silica gel plates (250 μm, Sorbent Technologies, Atlanta, GA, USA) and the plates were visualized under UV at 254 nm. Standard grade silica gel (230–400 mesh, Sorbent Technologies, Atlanta, GA, USA) was used for flash column chromatography. ^1^H and ^13^C nuclear magnetic resonance (NMR) spectra were recorded on a Bruker Avance III HD 600 MHz NMR or JEOL Model DELTA-270 (270 MHz) NMR spectrometer. Chemical shifts are reported in parts per million (δ) from an internal standard of residual DMSO-d_6_ (2.50 or 39.5 ppm) or CDCl_3_ (7.27 or 77.2 ppm). Data are reported as follows: chemical shift (δ), multiplicity (s, singlet; d, doublet; dd, doublet of doublet; t, triplet; dt, doublet of triplet, q, quartet; sep, septet; br s, broad singlet; m, multiplet), coupling constant (J) in Hertz (Hz), integration. High resolution mass spectrometry (HRMS) data were obtained on an Applied Biosystem 4000 Q TRAP^®^ LC/MS/MS system using electrospray ionization (ESI). Low-resolution mass spectra (LRMS) were recorded on a Shimadzu AXIMA Confidence MALDI-TOF mass spectrometer (nitrogen UV laser, 50 Hz, 337 nm) by using α-cyano-4-hydroxycinnamic acid (CHCA) as a matrix. Melting points were determined with a Mel-Temp^®^ capillary apparatus (Cole-Parmer, Staffordshire, UK) and are uncorrected. High performance liquid chromatography (HPLC) analyses were carried out on Agilent 1100 series HPLC system (Foster City, CA) equipped with a diode-array UV detector and a C18-bounded HPLC column (Vydac 218TP104, 4.6 × 250 mm, 10 μm) by using a 40 min-gradient elution from 10% to 90% acetonitrile in water (0.1% TFA) and a flow rate of 1.0 mL/min. Eluents were monitored at 280 nm. Solid-phase reactions were carried out in 12 mL polypropylene cartridges with 20 μ PE frit (Applied Separations, Allentown, PA) and a labquake tube shaker (Fisher Scientific, Pittsburgh, PA) was used for mixing.

### 4.1. Synthesis of Bis-Benzamides

*Methyl 3-isobutoxy-4-[(3-isobutoxybenzoyl)amino]benzoate* (**3**): Oxalyl chloride (0.26 mL, 3.0 mmol) was slowly added to a solution of 3-isobutoxybenzoic acid (300 mg, 1.5 mmol) in DCM (20 mL). The resulting mixture was stirred at room temperature for 2 h. The solvent and excess oxalyl chloride were then removed under reduced pressure, and the residue was dissolved in DCM (5 mL). The resulting solution was then slowly added to a solution of methyl 4-amino-3-isobutoxybenzoate (226 mg, 1.0 mmol) and DIEA (0.52 mL, 3.0 mmol) in DCM (20 mL). After stirring at room temperature for 12 h, the resulting solution was concentrated under reduced pressure, and diluted with EtOAc (20 mL) and 1N HCl (20 mL). The layers were separated, and the aqueous layer was extracted with EtOAc (20 mL). The organic layers were combined, washed with saturated NaHCO_3_ and brine, dried over anhydrous sodium sulfate, and concentrated under reduced pressure. The resulting solid was washed with EtOAc and dried in vacuo to afford the compound **3** as a white solid (240 mg, 59%). ^1^H NMR (DMSO-*d*_6_, 600 MHz): *δ* 9.44 (br s, 1 H), 8.12 (d, *J* = 8.3 Hz, 1 H), 7.63 (dd, *J* = 8.3, 1.7 Hz, 1 H), 7.56 (d, *J* = 1.7 Hz, 1 H), 7.49 (d, *J* = 7.7 Hz, 1 H), 7.46 (d, *J* = 7.7 Hz, 1 H), 7.45 (d, *J* = 7.7 Hz, 1 H), 7.19–7.17 (m, 1 H), 3.92 (d, *J* = 6.2 Hz, 2 H), 3.86 (s, 3 H), 3.83 (d, *J* = 6.6 Hz, 2 H), 2.14–2.01 (m, 2 H), 1.02 (d, *J* = 6.6 Hz, 6 H), 1.00 (d, *J* = 6.7 Hz, 6 H). ^13^C NMR (DMSO-*d*_6_, 150 MHz): *δ* 165.8, 164.6, 158.9, 149.5, 135.7, 131.8, 129.9, 125.9, 122.0, 121.9, 119.4, 118.5, 112.8, 112.0, 74.5, 73.9, 52.1, 27.8, 27.7, 18.99, 18.97. MALDI-TOF (*m*/*z*): [M+H]^+^ calcd for C_23_H_30_NO_5_: 400.21, found 400.81.

*Methyl 4-[(4-amino-3-isobutoxybenzoyl)amino]-3-isobutoxybenzoate* (**4**): The title compound was synthesized as previously described [10].

*Methyl 4-[(4-acetamido-3-isobutoxybenzoyl)amino]-3-isobutoxybenzoate* (**5a**): Acetyl chloride (0.015 mL, 0.21 mmol) was slowly added to a solution of compound **4** (60 mg, 0.14 mmol) and DIEA (0.097 mL, 0.56 mmol) in EtOAc (10 mL). After stirring at room temperature for 6 h, the resulting solution was concentrated under reduced pressure, and diluted with EtOAc (20 mL) and 1N HCl (10 mL). The layers were separated, and the aqueous layer was extracted with EtOAc (20 mL). The organic layers were combined, washed with saturated NaHCO_3_ and brine, dried over anhydrous sodium sulfate, and concentrated under reduced pressure. The crude product was purified by flash column chromatography (hexanes/DCM 1:1) to afford the compound **5a** as a light yellow solid (63 mg, 99%). Rf = 0.16 (hexanes/EtOAc 4:1). ^1^H NMR (DMSO-*d*_6_, 600 MHz): *δ* 9.37 (br s, 1 H), 9.10 (br s, 1 H), 8.15 (d, *J* = 8.4 Hz, 1 H), 8.12 (d, *J* = 8.3 Hz, 1 H), 7.63 (dd, *J* = 8.3, 1.5 Hz, 1 H), 7.56 (d, *J* = 1.5 Hz, 1 H), 7.54 (d, *J* = 1.5 Hz, 1 H), 7.51 (dd, *J* = 8.3, 1.5 Hz, 1 H), 3.92 (d, *J* = 6.6 Hz, 2 H), 3.89 (d, *J* = 6.6 Hz, 2 H), 3.86 (s, 3 H), 2.19–2.11 (m, 2 H), 2.15 (s, 3 H), 1.03 (d, *J* = 7.0 Hz, 12 H). ^13^C NMR (DMSO-*d*_6_, 150 MHz): *δ* 168.8, 165.8, 164.2, 149.3, 148.4, 131.9, 131.1, 129.5, 125.6, 122.1, 121.6, 121.1, 119.8, 112.0, 110.8, 74.8, −74.6, 52.1, 27.8, 27.5, 24.0, 19.10, 19.05. MALDI-TOF (*m*/*z*): [M + Na]^+^ calcd for C_25_H_32_N_2_NaO_6_: 479.52, found 479.79.

Methyl 3-isobutoxy-4-[[3-isobutoxy-4-(pentanoylamino)benzoyl]amino]benzoate (**5b**): The title compound was prepared following the same procedure as for compound **5a**, using compound **4** (250 mg, 0.60 mmol) and valeroyl chloride (0.15 mL, 1.21 mmol). The crude product was purified by flash column chromatography (hexanes/EtOAc 2:1) to afford the compound **5b** as a white solid (189 mg, 63%). Rf = 0.49 (hexanes/EtOAc 2:1). ^1^H NMR (DMSO-*d*_6_, 600 MHz): *δ* 9.37 (br s, 1 H), 9.02 (br s, 1 H), 8.15 (d, *J* = 8.4 Hz, 1 H), 8.08 (d, *J* = 8.4 Hz, 1 H), 7.64 (dd, *J* = 8.3, 1.7 Hz, 1 H), 7.56 (d, *J* = 1.8 Hz, 1 H), 7.54 (d, *J* = 1.8 Hz, 1 H), 7.51 (dd, *J* = 8.4, 1.8 Hz, 1 H), 3.92 (d, *J* = 6.2 Hz, 2 H), 3.89 (d, *J* = 6.6 Hz, 2 H), 3.86 (s, 3 H), 2.44 (t, *J* = 7.3 Hz, 2 H), 2.17–2.09 (m, 2 H), 1.61–1.56 (m, 2 H), 1.37–1.31 (m, 2 H), 1.03 (d, *J* = 6.6 Hz, 12 H), 0.90 (t, *J* = 7.3 Hz, 3 H). ^13^C NMR (DMSO-*d*_6_, 150 MHz): *δ* 171.6, 165.8, 164.2, 149.3, 148.6, 131.9, 131.0, 129.6, 125.6, 122.1, 121.5, 121.3, 119.8, 111.9, 110.7, 74.7, 74.5, 52.1, 36.0, 27.8, 27.6, 27.3, 21.7, 19.1, 13.7. MALDI-TOF (*m*/*z*): [M + H]^+^ calcd for C_28_H_39_N_2_O_6_: 499.28, found 499.82.

*4-[2-Isobutoxy-4-[(2-isobutoxy-4-methoxycarbonylphenyl)carbamoyl]anilino]-4-oxobutanoic acid* (**5c**): Succinic anhydride (3.4 g, 33.8 mmol) was added to a solution of compound **4** (2.8 g, 6.8 mmol) in DCM (300 mL). After stirring at room temperature for 12 h, the resulting precipitate was collected by vacuum filtration, washed with DCM, and dried in vacuo to afford compound **5c** as a white solid (2.4 g, 69%). ^1^H NMR (DMSO-*d*_6_, 600 MHz): *δ* 12.14 (br s, 1 H), 9.37 (br s, 1 H), 9.12 (br s, 1 H), 8.15 (d, *J* = 8.1 Hz, 1 H), 8.13 (d, *J* = 7.4 Hz, 1 H), 7.63 (dd, *J* = 8.3, 1.7 Hz, 1 H), 7.56 (d, *J* = 1.7 Hz, 1 H), 7.54 (d, *J* = 1.7 Hz, 1 H), 7.51 (dd, *J* = 8.3, 1.7 Hz, 1 H), 3.92 (d, *J* = 6.2 Hz, 2 H), 3.89 (d, *J* = 6.6 Hz, 2 H), 3.86 (s, 3 H), 2.69 (t, *J* = 6.7 Hz, 2 H), 2.53 (t, *J* = 6.7 Hz, 2 H), 2.19–2.10 (m, 2 H), 1.04 (d, *J* = 6.7 Hz, 6 H), 1.03 (d, *J* = 6.7 Hz, 6 H). ^13^C NMR (DMSO-*d*_6_, 150 MHz): *δ* 173.8, 170.6, 165.8, 164.2, 149.2, 148.3, 131.9, 131.1, 129.4, 125.6, 122.1, 121.5, 120.7, 119.9, 111.9, 110.7, 74.8, 74.5, 52.1, 31.3, 28.9, 27.8, 27.6, 19.12, 19.05. MALDI-TOF (*m*/*z*): [M + H]^+^ calcd for C_27_H_35_N_2_O_8_: 515.24, found 515.69.

[2-[2-Isobutoxy-4-[(2-isobutoxy-4-methoxycarbonylphenyl)carbamoyl]anilino]-2-oxoethyl]ammonium trifluoroacetate (**5e**): Compound **5d** was synthesized as previously described [10]. Then, compound **5d** (1.8 g, 3.2 mmol) was dissolved in 50% TFA in DCM (40 mL) and the mixture was stirred at room temperature for 1 h. The reaction mixture was concentrated under reduced pressure. The product was precipitated by adding cold diethyl ether, washed with diethyl ether, and dried in vacuo. The TFA salt of compound **5e** was obtained as a white solid (1.7 g, 93%). ^1^H NMR (DMSO-*d*_6_, 600 MHz): *δ* 9.73 (br s, 1 H), 9.43 (br s, 1 H), 8.15 (d, *J* = 8.1 Hz, 1 H), 8.13 (d, *J* = 8.4 Hz, 1 H), 8.00 (br s, 3 H), 7.64 (dd, *J* = 8.4, 1.5 Hz, 1 H), 7.60 (d, *J* = 1.5 Hz, 1 H), 7.57–7.56 (m, 2 H), 3.92 (d, *J* = 5.5 Hz, 4 H), 3.91 (s, 2 H), 3.86 (s, 3 H), 1.04 (d, *J* = 5.8 Hz, 6 H), 1.03 (d, *J* = 6.2 Hz, 6 H). ^13^C NMR (DMSO-*d*_6_, 150 MHz): *δ* 165.8, 164.2, 158.4, 158.2, 149.4, 148.7, 131.9, 130.4, 129.9, 125.8, 122.1, 121.8, 121.3, 120.0, 112.0, 111.0, 75.0, 74.6, 52.1, 41.4, 27.8, 27.5, 19.13, 19.06. MALDI-TOF (*m*/*z*): [M + H]^+^ calcd for C_25_H_34_N_3_O_6_: 472.24, found 472.67.

*3-Isobutoxy-N-(2-isobutoxyphenyl)-4-nitrobenzamide* (**7b**): The title compound was prepared following the same procedure as for compound **3**, using 3-isobutoxy-4-nitrobenzoic acid (431 mg, 0.18 mmol) and 2-isobutoxyaniline (**6b**) (200 mg, 1.2 mmol). Recrystallization from EtOAc/hexanes afforded compound **7b** as a light yellow solid (210 mg, 45%). ^1^H NMR (DMSO-*d*_6_, 600 MHz): *δ* 9.70 (br s, 1 H), 8.01 (d, *J* = 8.1 Hz, 1 H), 7.80 (s, 1 H), 7.66 (d, *J* = 8.1 Hz, 1 H), 7.60 (d, *J* = 8.4 Hz, 1 H), 7.23–7.20 (m, 1 H), 7.10–7.09 (m, 1 H), 6.99–6.97 (m, 1 H), 4.02 (d, *J* = 6.2 Hz, 2 H), 3.80 (d, *J* = 6.2 Hz, 2 H), 2.10–2.00 (m, 2 H), 0.99 (d, *J* = 6.6 Hz, 6 H), 0.97 (d, *J* = 6.6 Hz, 6 H). ^13^C NMR (DMSO-*d*_6_, 150 MHz): *δ* 163.2, 151.5, 150.9, 140.8, 139.5, 126.4, 126.2, 125.0, 124.9, 120.0, 119.3, 113.6, 112.4, 75.1, 75.0, 74.1, 27.7, 27.5, 18.9, 18.6. MALDI-TOF (*m*/*z*): [M + H]^+^ calcd for C_21_H_27_N_2_O_5_: 387.19, found 387.67.

*3-Isobutoxy-4-[(3-isobutoxy-4-nitrobenzoyl)amino]benzoic acid* (**8**): Compound **7a** was synthesized as previously described [10]. Then, Pd(PPh_3_)_4_ (198 mg, 0.17 mmol) and PhSiH_3_ (0.43 mL, 3.4 mmol) were added to a solution of compound **7a** (800 mg, 1.7 mmol) in DCM (30 mL). After stirring at room temperature for 1 h, the reaction mixture was concentrated under reduced pressure. The product was precipitated by adding diethyl ether, washed with diethyl ether, and dried in vacuo. Compound **8** was obtained as a white solid (465 mg, 64%), m.p. 242–245 °C. ^1^H NMR (DMSO-*d*_6_, 600 MHz): *δ* 12.98 (br s, 1 H), 9.78 (br s, 1 H), 8.03 (d, *J* = 8.4 Hz, 1 H), 7.96 (d, *J* = 8.4 Hz, 1 H), 7.80 (d, *J* = 1.4 Hz, 1 H), 7.62–7.60 (m, 2 H), 7.57 (d, *J* = 1.5 Hz, 1 H), 4.03 (d, *J* = 6.2 Hz, 2 H), 3.89 (d, *J* = 6.6 Hz, 2 H), 2.11–2.05 (m, 2 H), 1.005 (d, *J* = 6.4 Hz, 6 H), 0.995 (d, *J* = 6.3 Hz, 6 H). ^13^C NMR (DMSO-*d*_6_, 150 MHz): *δ* 167.4, 164.0, 151.6, 150.8, 141.7, 139.8, 131.3, 128.6, 125.6, 123.6, 122.4, 120.0, 114.3, 113.0, 75.8, 75.0, 40.2, 28.3, 28.1, 19.5, 19.2. MALDI-TOF (*m*/*z*): [M + Na]^+^ calcd for C_22_H_26_N_2_NaO_7_: 453.16, found 453.67.

*3-Isobutoxy-4-[(3-isobutoxy-4-nitrobenzoyl)amino]benzamide* (**9a**): DIEA (0.16 mL, 0.92 mmol) was added to a solution of compound **8** (200 mg, 0.46 mmol) and PyBOP (263 mg, 0.51 mmol) in DMF (10 mL). After stirring at room temperature for 1 h, ammonium chloride (246 mg, 4.6 mmol) and additional DIEA (0.80 mL, 4.6 mmol) were added. The resulting mixture was stirred at room temperature for 12 h and diluted with EtOAc (50 mL) and 1N HCl (30 mL). The layers were separated, and the aqueous layer was extracted with EtOAc (30 mL). The organic layers were combined, washed with saturated NaHCO_3_, and concentrated under reduced pressure. The resulting solid was washed with EtOAc and dried in vacuo to afford the compound **9a** as a white solid (175 mg, 89%), m.p. 249–251 °C. ^1^H NMR (DMSO-*d*_6_, 270 MHz): *δ* 9.76 (br s, 1 H), 8.02 (d, *J* = 8.4 Hz, 1 H), 8.00 (br s, 1 H), 7.84 (d, *J* = 8.2 Hz, 1 H), 7.79 (d, *J* = 1.2 Hz, 1 H), 7.60 (dd, *J* = 8.4, 1.2 Hz, 1 H), 7.57 (d, *J* = 1.5 Hz, 1 H), 7.53 (dd, *J* = 8.2, 1.5 Hz, 1 H), 7.37 (br s, 1 H), 4.02 (d, *J* = 6.7 Hz, 2 H), 3.87 (d, *J* = 6.4 Hz, 2 H), 2.12–2.00 (m, 2 H), 0.994 (d, *J* = 6.7 Hz, 6 H), 0.992 (d, *J* = 6.7 Hz, 6 H). ^13^C NMR (DMSO-*d*_6_, 150 MHz): *δ* 167.2, 163.5, 151.0, 150.6, 141.1, 139.4, 131.9, 129.2, 125.1, 123.5, 119.8, 119.5, 113.8, 111.4, 75.3, 74.5, 27.8, 27.6, 19.1, 18.7. MALDI-TOF (*m*/*z*): [M + Na]^+^ calcd for C_22_H_27_N_3_NaO_6_: 452.18, found 452.54.

*3-Isobutoxy-4-[(3-isobutoxy-4-nitrobenzoyl)amino]-N-isobutylbenzamide* (**9b**): DIEA (0.084 mL 0.48 mmol) was added to a mixture of compound **8** (50 mg, 0.12 mmol) and PyBOP (73 mg, 0.14 mmol) in DMF (5 mL), followed by isobutylamine (0.024 mL, 0.24 mmol). After stirring at room temperature for 12 h, the resulting solution was diluted with EtOAc (20 mL) and 1N HCl (20 mL). The layers were separated, and the aqueous layer was extracted with EtOAc (20 mL). The organic layers were combined, washed with saturated NaHCO_3_ and brine, dried over anhydrous sodium sulfate, and concentrated under reduced pressure. The crude product was purified by flash column chromatography (hexanes/EtOAc 1:1) to afford the compound **9b** as a light yellow solid (47 mg, 81%). Rf = 0.41 (hexanes/EtOAc 1:1). ^1^H NMR (DMSO-*d*_6_, 600 MHz): *δ* 9.77 (br s, 1 H), 8.47 (d, *J* = 5.7 Hz, 1 H), 8.02 (d, *J* = 8.2 Hz, 1 H), 7.84 (d, *J* = 8.2 Hz, 1 H), 7.80 (br s, 1 H), 7.61 (dd, *J* = 8.2, 1.4 Hz, 1 H), 7.54 (br s, 1 H), 7.51 (d, *J* = 8.2 Hz, 1 H), 4.03 (d, *J* = 6.2 Hz, 2 H), 3.88 (d, *J* = 6.2 Hz, 2 H), 3.09 (t, *J* = 6.4 Hz, 2 H), 2.11–2.04 (m, 2 H), 1.90–1.82 (m, 1 H), 1.00 (d, *J* = 5.7 Hz, 6 H), 0.996 (d, *J* = 5.7 Hz, 6 H), 0.90 (d, *J* = 6.6 Hz, 6 H). ^13^C NMR (DMSO-*d*_6_, 150 MHz): *δ* 165.5, 163.4, 151.0, 150.7, 141.1, 139.4, 132.4, 128.9, 125.1, 123.7, 119.5, 119.4, 113.8, 111.1, 75.2, 74.5, 46.8, 28.1, 27.8, 27.6, 20.2, 19.1, 18.7. MALDI-TOF (*m*/*z*): [M + H]^+^ calcd for C_26_H_36_N_3_O_6_: 486.26, found 486.76.

2-[[3-Isobutoxy-4-[(3-isobutoxy-4-nitrobenzoyl)amino]benzoyl]amino]acetic acid (**9d**): DIEA (0.35 mL, 2.0 mmol) was added to a solution of compound **8** (430 mg, 1.0 mmol) and PyBOP (624 mg, 1.2 mmol) in DMF (20 mL). After stirring at room temperature for 1 h, glycine allyl ester trifluoroacetate (344 mg, 1.5 mmol) and additional DIEA (0.35 mL, 2.0 mmol) were added. The resulting mixture was stirred at room temperature for 12 h and diluted with EtOAc (50 mL) and 1N HCl (30 mL). The layers were separated, and the aqueous layer was extracted with EtOAc (20 mL). The organic layers were combined, washed with saturated NaHCO_3_ and brine, dried over anhydrous sodium sulfate, and concentrated under reduced pressure. Recrystallization from EtOAc/hexanes afforded compound **9c** as a white solid (308 mg, 58%). Then, Pd(PPh_3_)_4_ (54 mg, 0.047 mmol) and PhSiH_3_ (0.12mL, 0.94 mmol) were added to a solution of compound **9c** (250 mg, 0.47 mmol) in THF (20 mL). After stirring at room temperature for 1 h, the reaction mixture was concentrated under reduced pressure. The product was precipitated by adding diethyl ether, washed with diethyl ether, and dried in vacuo. Compound **9d** was obtained as a white solid (165 mg, 72%). ^1^H NMR (DMSO-*d*_6_, 600 MHz): *δ* 12.6 (br s, 1 H), 9.79 (br s, 1 H), 8.86 (t, *J* = 5.7 Hz, 1 H), 8.02 (d, *J* = 8.4 Hz, 1 H), 7.88 (d, *J* = 8.1 Hz, 1 H), 7.81 (br s, 1 H), 7.62 (d, *J* = 8.4 Hz, 1 H), 7.58 (br s, 1 H), 7.54 (d, *J* = 8.2 Hz, 1 H), 4.03 (d, *J* = 6.6 Hz, 2 H), 3.94 (d, *J* = 5.9 Hz, 2 H), 3.89 (d, *J* = 6.2 Hz, 2 H), 2.12–2.05 (m, 2 H), 1.01 (d, *J* = 6.6 Hz, 6 H), 1.00 (d, *J* = 6.7 Hz, 6 H). ^13^C NMR (DMSO-*d*_6_, 150 MHz): *δ* 171.4, 165.8, 163.5, 151.1, 150.7, 141.1, 139.4, 131.4, 129.4, 125.1, 123.6, 119.7, 119.5, 113.8, 111.1, 75.3, 74.6, 41.3, 27.8, 27.7, 19.1, 18.7. MALDI-TOF (*m*/*z*): [M + Na]^+^ calcd for C_24_H_29_N_3_NaO_8_: 510.19, found 510.68.

2-[[3-Isobutoxy-4-[(3-isobutoxy-4-nitrobenzoyl)amino]benzoyl]amino]ethylammonium trifluoroacetate (**9f**): The title compound was synthesized as previously described [10].

### 4.2. Synthesis of Bis-Benzamides Library 14

These compounds were synthesized as previously described [21]. We briefly describe their syntheses. Fmoc-Rink amide MBHA resin (0.50 mmol/g, 300 mg, 0.15 mmol) was swollen in DMF for 2 h and washed with DMF (×3). The Fmoc protecting group was removed by treating with piperidine (20% in DMF, 5 × 30 min), and washed with DMF (×3). Then, 3-alkoxy-4-nitrobenzoic acid **10** was introduced by using a preactivated ester which was prepared by mixing 3-alkoxy-4-nitrobenzoic acid **10** (4.0 equiv.), PyBOP (4.0 equiv), and DIEA (4.0 equiv) in DMF (8 mL) for 5 min. The solution was added to the resin and shaken at room temperature for 24 h. The resin was then filtered and washed with DMF (×3) affording compound **11**. The nitro group of the compound **11** was swollen in AcOH (50% in H_2_O)/HCl (0.5 N in H_2_O)/THF (1:1:6, 8 mL) for 20 min and treated with SnCl_2_·2H_2_O (5.0 equiv.). The reaction mixture was shaken at room temperature for 24 h. The resin was filtered, and washed with HCl (0.5 N in H_2_O)/DMF (1:6) (×3), H_2_O/DMF (1:6) (×3) and DMF (×3) affording compound **12**. Then 3-alkoxy-4-nitrobenzoic acid **10** was introduced by using a preactivated HOAt ester which was prepared by mixing 3-alkoxy-4-nitrobenzoic acid **10** (4.0 equiv.), HATU (4.0 equiv), and DIEA (4.0 equiv) in DMF (8 mL) for 1 h. The solution was added to the resin and shaken at room temperature for 24 h. The resin was then filtered and washed with DMF (×3) affording the resin-bound bis-benzamide **13**. The bis-benzamide **13** was washed with DCM (×3) and dried in vacuo. The dried resin was treated with a cleavage mixture of TFA/H_2_O (95:5, 6 mL) for 90 min. The TFA solution was then filtered, and the resin was washed with TFA (2 mL). The combined TFA solution was concentrated to a volume of approximately 0.5 mL with a gentle stream of nitrogen. The product was precipitated by adding cold diethyl ether, washed with diethyl ether, and dried in vacuo affording compound **14**.

*4-[(4-Nitro-3-propoxybenzoyl)amino]-3-propoxybenzamide* (**14a**): Light yellow solid, 22 mg, 47% overall yield, 95% purity by HPLC. ^1^H NMR (DMSO-*d*_6_, 270 MHz): *δ* 9.76 (br s, 1 H), 8.01 (d, *J* = 8.4 Hz, 1 H), 7.99 (br s, 1 H), 7.85 (d, *J* = 8.2 Hz, 1 H), 7.80 (d, *J* = 1.4 Hz, 1 H), 7.61 (dd, *J* = 8.4, 1.5 Hz, 1 H), 7.58 (d, *J* = 1.6 Hz, 1 H), 7.53 (dd, *J* = 8.0, 1.6 Hz, 1 H), 7.37 (br s, 1 H), 4.22 (t, *J* = 6.4 Hz, 2 H), 4.05 (t, *J* = 6.4 Hz, 2 H), 1.82–1.73 (m, 4 H), 0.988 (t, *J* = 7.2 Hz, 3 H), 0.986 (t, *J* = 7.2 Hz, 3 H). ^13^C NMR (DMSO-*d*_6_, 150 MHz): *δ* 167.2, 163.5, 151.0, 150.4, 141.2, 139.4, 131.8, 129.2, 125.0, 123.4, 119.7, 119.5, 114.0, 111.4, 70.8, 69.9, 22.0, 21.8, 10.4, 10.2. MALDI-TOF (*m*/*z*): [M + Na]^+^ calcd for C_20_H_23_N_3_NaO_6_: 424.15, found 424.87. 

*4-[(3-Isopropoxy-4-nitrobenzoyl)amino]-3-propoxybenzamide* (**14b**): Light yellow solid, 22 mg, 37% overall yield, 97% purity by HPLC. ^1^H NMR (DMSO-*d*_6_, 270 MHz): *δ* 9.74 (br s, 1 H), 7.99 (br s, 1 H), 7.97 (d, *J* = 8.4 Hz, 1 H), 7.86 (d, *J* = 8.2 Hz, 1 H), 7.85 (br s, 1 H), 7.81 (d, *J* = 8.3, 1 H), 7.57 (br s, 1 H), 7.53 (d, *J* = 8.4, 1 H), 7.37 (br s, 1 H), 4.94 (sep, *J* = 6.2, 1 H), 4.05 (t, *J* = 6.3 Hz, 2 H), 1.82–1.71 (m, 2 H), 1.33 (d, *J* = 6.2, 6 H), 0.99 (t, *J* = 7.3 Hz, 3 H). ^13^C NMR (DMSO-*d*_6_, 150 MHz): *δ* 167.2, 163.6, 150.4, 149.7, 142.3, 139.2, 131.7, 129.2, 124.9, 123.3, 119.7, 119.6, 115.1, 111.4, 72.5, 69.9, 22.0, 21.6, 10.4. MALDI-TOF (*m*/*z*): [M + Na]^+^ calcd for C_20_H_23_N_3_NaO_6_: 424.15, found 424.97.

*4-[(3-Butoxy-4-nitrobenzoyl)amino]-3-propoxybenzamide* (**14c**): Light yellow solid, 30 mg, 48% overall yield based on the loading of Fmoc-Rink amide resin, 95% purity by HPLC, m.p. 247–248 °C. ^1^H NMR (DMSO-*d*_6_, 270 MHz): *δ* 9.76 (br s, 1 H), 8.00 (d, *J* = 8.2 Hz, 1 H), 7.99 (br s, 1 H), 7.85 (d, *J* = 8.2 Hz, 1 H), 7.81 (d, *J* = 1.5 Hz, 1 H), 7.60 (dd, *J* = 8.4, 1.5 Hz, 1 H), 7.58 (d, *J* = 1.7 Hz, 1 H), 7.53 (dd, *J* = 8.2, 1.7 Hz, 1 H), 7.37 (br s, 1 H), 4.26 (t, *J* = 6.3 Hz, 2 H), 4.05 (t, *J* = 6.3 Hz, 2 H), 1.85–1.69 (m, 4 H), 1.51–1.38 (m, 2 H), 0.99 (t, *J* = 7.4 Hz, 3 H), 0.94 (t, *J* = 7.7 Hz, 3 H). ^1^^3^C NMR (DMSO-*d*_6_, 68 MHz): *δ* 167.8, 164.1, 151.5, 150.9, 141.8, 140.0, 132.3, 129.8, 125.6, 123.9, 120.3, 120.0, 114.5, 112.0, 70.5, 70.0, 30.9, 22.6, 19.1, 14.1, 11.0. HRMS-ESI (*m*/*z*): [M + H]^+^ calcd for C_21_H_26_N_3_O_6_: 416.1822, found 416.1815.

*4-[(3-Isobutoxy-4-nitrobenzoyl)amino]-3-propoxybenzamide* (**14d**): Light yellow solid, 32 mg, 51% overall yield based on the loading of Fmoc-Rink amide resin, 93% purity by HPLC, m.p. 255–257 °C. ^1^H NMR (DMSO-*d*_6_, 270 MHz): *δ* 9.76 (br s, 1 H), 8.02 (d, *J* = 8.2 Hz, 1 H), 7.99 (br s, 1 H), 7.86 (d, *J* = 8.2 Hz, 1 H), 7.79 (d, *J* = 1.6 Hz, 1 H), 7.61 (dd, *J* = 8.3, 1.6 Hz, 1 H), 7.58 (d, *J* = 1.7 Hz, 1 H), 7.53 (dd, *J* = 8.2, 1.7 Hz, 1 H), 7.37 (br s, 1 H), 4.05 (t, *J* = 6.4 Hz, 2 H), 4.03 (d, *J* = 6.5 Hz, 2 H), 2.12–2.02 (m, 1 H), 1.85–1.72 (m, 2 H), 0.994 (t, *J* = 7.2 Hz, 3 H), 0.992 (d, *J* = 6.7 Hz, 6 H). ^1^^3^C NMR (DMSO-*d*_6_, 68 MHz): *δ* 167.8, 164.1, 151.6, 151.0, 141.7, 140.0, 132.3, 129.8, 125.6, 123.9, 120.3, 120.1, 114.5, 112.0, 75.8, 70.5, 28.2, 22.6, 19.3, 11.2. HRMS-ESI (*m*/*z*): [M + H]^+^ calcd for C_21_H_26_N_3_O_6_: 416.1816, found 416.1822.

*3-Isopropoxy-4-[(4-nitro-3-propoxy-benzoyl)amino]benzamide* (**14f**): Light yellow solid, 24 mg, 40% overall yield, 95% purity by HPLC. ^1^H NMR (DMSO-*d*_6_, 270 MHz): *δ* 9.66 (br s, 1 H), 8.01(d, *J* = 8.4 Hz, 1 H), 7.98 (br s, 1 H), 7.91 (d, *J* = 8.2 Hz, 1 H), 7.78 (d, *J* = 1.4 Hz, 1 H), 7.60 (dd, *J* = 8.2, 1.4 Hz, 1 H), 7.58 (d, *J* = 1.5 Hz, 1 H), 7.52 (dd, *J* = 8.4, 1.6 Hz, 1 H), 7.37 (br s, 1 H), 4.70 (sep, *J* = 6.2 Hz, 1 H), 4.23 (t, *J* = 6.4 Hz, 2 H), 1.83–1.71 (m, 2 H), 1.32 (d, *J* = 5.7 Hz, 6 H), 0.99 (t, *J* = 7.4 Hz, 3 H). ^13^C NMR (DMSO-*d*_6_, 150 MHz): *δ* 167.2, 163.6, 151.0, 148.9, 141.2, 139.5, 131.5, 130.2, 125.1, 123.1, 119.9, 119.4, 114.1, 113.1, 71.2, 70.8, 21.8, 21.7, 10.2. MALDI-TOF (*m*/*z*): [M + Na]^+^ calcd for C_20_H_23_N_3_NaO_6_: 424.15, found 424.62.

*4-[(3-Butoxy-4-nitrobenzoyl)amino]-3-isopropoxybenzamide* (**14h**): Light yellow solid, 28 mg, 45% overall yield based on the loading of Fmoc-Rink amide resin, 96% purity by HPLC, m.p. 213–214 °C. ^1^H NMR (DMSO-*d*_6_, 270 MHz): *δ* 9.65 (br s, 1 H), 8.01 (d, *J* = 8.4 Hz, 1 H), 7.99 (br s, 1 H), 7.92 (d, *J* = 8.4 Hz, 1 H), 7.79 (br s, 1 H), 7.59 (d, *J* = 8.3 Hz, 1 H), 7.58 (br s, 1 H), 7.52 (d, *J* = 8.4 Hz, 1 H), 7.37 (br s, 1 H), 4.70 (sep, *J* = 5.9 Hz, 1 H), 4.27 (t, *J* = 6.4 Hz, 2 H), 1.79–1.68 (m, 2 H), 1.51–1.37 (m, 2 H), 1.32 (d, *J* = 5.9 Hz, 6 H), 0.93 (t, *J* = 7.4 Hz, 3 H). ^13^C NMR (DMSO-*d*_6_, 150 MHz): *δ* 167.2, 163.5, 150.9, 148.9, 141.2, 139.4, 131.5, 130.2, 125.0, 122.9, 119.9, 119.4, 114.0, 113.0, 71.2, 69.1, 30.3, 21.8, 18.5, 13.5. MALDI-TOF (*m*/*z*): [M + Na]^+^ calcd for C_21_H_25_N_3_NaO_6_: 438.16, found 438.51.

*4-[(3-Isobutoxy-4-nitrobenzoyl)amino]-3-isopropoxybenzamide* (**14i**): Light yellow solid, 31 mg, 48% overall yield, 93% purity by HPLC. ^1^H NMR (DMSO-*d*_6_, 270 MHz): *δ* 9.66 (br s, 1 H), 8.02 (d, *J* = 8.4 Hz, 1 H), 7.99 (br s, 1 H), 7.92 (d, *J* = 8.2 Hz, 1 H), 7.77 (br s, 1 H), 7.59 (d, *J* = 8.3 Hz, 1 H), 7.58 (br s, 1 H), 7.52 (d, *J* = 8.4 Hz, 1 H), 7.36 (br s, 1 H), 4.70 (sep, *J* = 5.9 Hz, 1 H), 4.04 (d, *J* = 6.4 Hz, 2 H), 2.11–2.02 (m, 1 H), 1.32 (d, *J* = 5.7 Hz, 6 H), 0.99 (d, *J* = 6.7 Hz, 6 H). ^13^C NMR (DMSO-*d*_6_, 150 MHz): *δ* 167.2, 163.5, 151.0, 148.9, 141.1, 139.5, 131.5, 130.2, 125.1, 123.0, 119.9, 119.4, 114.0, 113.0, 75.2, 71.2, 27.6, 21.8, 18.7. MALDI-TOF (*m*/*z*): [M + Na]^+^ calcd for C_21_H_25_N_3_NaO_6_: 438.16, found 438.95.

*3-Butoxy-4-[(4-nitro-3-propoxybenzoyl)amino]benzamide* (**14k**): Light yellow solid, 23 mg, 37% overall yield, >99% purity by HPLC. ^1^H NMR (DMSO-*d*_6_, 270 MHz): *δ* 9.75 (br s, 1 H), 8.01 (d, *J* = 8.4 Hz, 1 H), 7.99 (br s, 1 H), 7.85 (d, *J* = 8.2 Hz, 1 H), 7.80 (d, *J* = 1.2 Hz, 1 H), 7.60 (dd, *J* = 8.4, 1.4 Hz, 1 H), 7.58 (d, *J* = 1.4 Hz, 1 H), 7.52 (dd, *J* = 8.4, 1.5 Hz, 1 H), 7.37 (br s, 1 H), 4.22 (t, *J* = 6.4 Hz, 2 H), 4.09 (t, *J* = 6.4 Hz, 2 H), 1.84–1.70 (m, 4 H), 1.52–1.38 (m, 2 H), 0.99 (t, *J* = 7.4 Hz, 3 H), 0.91 (t, *J* = 7.4 Hz, 3 H). ^13^C NMR (DMSO-*d*_6_, 150 MHz): *δ* 167.2, 163.5, 150.9, 150.4, 141.2, 139.4, 131.8, 129.2, 125.0, 123.3, 119.8, 119.5, 114.0, 111.4, 70.8, 68.1, 30.7, 21.8, 18.7, 13.7, 10.2. MALDI-TOF (*m*/*z*): [M + Na]^+^ calcd for C_21_H_25_N_3_NaO_6_: 438.16, found 438.75.

*3-Butoxy-4-[(3-isopropoxy-4-nitrobenzoyl)amino]benzamide* (**14l**): Light yellow solid, 25 mg, 40% overall yield based on the loading of Fmoc-Rink amide resin, >99% purity by HPLC, m.p. 233–235 °C. ^1^H NMR (DMSO-*d*_6_, 270 MHz): *δ* 9.73 (br s, 1 H), 7.99 (br s, 1 H), 7.97 (d, *J* = 8.4 Hz, 1 H), 7.86 (d, *J* = 8.2 Hz, 1 H), 7.81 (d, *J* = 1.2 Hz, 1 H), 7.59 (dd, *J* = 8.2, 1.2 Hz, 1 H), 7.58 (d, *J* = 1.4 Hz, 1 H), 7.52 (dd, *J* = 8.3, 1.4 Hz, 1 H), 7.37 (br s, 1 H), 4.94 (sep, *J* = 5.9 Hz, 1 H), 4.09 (t, *J* = 6.2 Hz, 2 H), 1.80–1.70 (m, 2 H), 1.52–1.38 (m, 2 H), 1.33 (d, *J* = 6.2 Hz, 6 H), 0.91 (t, *J* = 7.4 Hz, 3 H). ^13^C NMR (DMSO-*d*_6_, 150 MHz): *δ* 167.2, 163.5, 150.4, 149.7, 142.3, 139.2, 131.7, 129.2, 124.9, 123.3, 119.7, 119.6, 115.1, 111.4, 72.5, 68.2, 30.7, 21.5, 18.7, 13.7. MALDI-TOF (*m*/*z*): [M + H]^+^ calcd for C_21_H_26_N_3_O_6_: 416.18, found 416.56.

*3-Butoxy-4-[(3-butoxy-4-nitrobenzoyl)amino]benzamide* (**14m**): Light yellow solid, 20 mg, 31% overall yield based on the loading of Fmoc-Rink amide resin, 92% purity by HPLC, m.p. 249–251 °C. ^1^H NMR (DMSO-*d*_6_, 270 MHz): *δ* 9.74 (br s, 1 H), 8.00 (d, *J* = 8.4 Hz, 1 H), 7.99 (br s, 1 H), 7.86 (d, *J* = 8.4 Hz, 1 H), 7.80 (d, *J* = 1.2 Hz, 1 H), 7.60 (dd, *J* = 8.4, 1.4 Hz, 1 H), 7.58 (br s, 1 H), 7.52 (dd, *J* = 8.3, 1.4 Hz, 1 H), 7.37 (br s, 1 H), 4.26 (t, *J* = 6.2 Hz, 2 H), 4.09 (t, *J* = 6.2 Hz, 2 H), 1.80–1.69 (m, 4 H), 1.52–1.37 (m, 4 H), 0.94 (t, *J* = 7.2 Hz, 3 H), 0.91 (t, *J* = 7.3 Hz, 3 H). ^1^^3^C NMR (DMSO-*d*_6_, 68 MHz): *δ* 167.8, 164.1, 151.5, 151.0, 141.8, 140.0, 132.3, 129.8, 125.6, 123.9, 120.3, 120.1, 114.5, 112.0, 69.7, 68.7, 31.2, 30.9, 19.3, 19.1, 14.3, 14.1. MALDI-TOF (*m*/*z*): [M + Na]^+^ calcd for C_22_H_27_N_3_NaO_6_: 452.18, found 452.56.

*3-Butoxy-4-[(3-sec-butoxy-4-nitrobenzoyl)amino]benzamide* (**14o**): Light yellow solid, 20 mg, 31% overall yield based on the loading of Fmoc-Rink amide resin, 94% purity by HPLC, m.p. 218–220 °C. ^1^H NMR (DMSO-*d*_6_, 270 MHz): *δ* 9.73 (br s, 1 H), 7.99 (br s, 1 H), 7.97 (d, *J* = 8.4 Hz, 1 H), 7.87 (d, *J* = 8.2 Hz, 1 H), 7.80 (d, *J* = 1.2 Hz, 1 H), 7.580 (d, *J* = 1.4 Hz, 1 H), 7.578 (dd, *J* = 8.2, 1.5 Hz, 1 H), 7.52 (dd, *J* = 8.3, 1.6 Hz, 1 H), 7.37 (br s, 1 H), 4.81–4.70 (m, 1 H), 4.09 (d, *J* = 6.4 Hz, 2 H), 1.80–1.63 (m, 4 H), 1.52–1.38 (m, 2 H), 1.30 (d, *J* = 5.9 Hz, 3 H), 0.93 (t, *J* = 7.4 Hz, 3 H), 0.90 (t, *J* = 7.4 Hz, 3 H). ^1^^3^C NMR (DMSO-*d*_6_, 150 MHz): *δ* 167.2, 163.6, 150.4, 150.0, 142.2, 139.2, 131.7, 129.2, 124.9, 123.3, 119.8, 119.5, 114.8, 111.4, 77.0, 68.2, 30.7, 28.4, 18.7, 18.7, 13.7, 9.2. MALDI-TOF (*m*/*z*): [M + Na]^+^ calcd for C_22_H_27_N_3_NaO_6_: 452.18, found 452.60.

*4-[(3-Butoxy-4-nitrobenzoyl)amino]-3-isobutoxybenzamide* (**14r**): Light yellow solid, 20 mg, 31% overall yield based on the loading of Fmoc-Rink amide resin, 92% purity by HPLC, m.p. 239–240 °C. ^1^H NMR (DMSO-*d*_6_, 270 MHz): *δ* 9.76 (br s, 1 H), 8.01 (d, *J* = 8.2 Hz, 1 H), 7.99 (br s, 1 H), 7.83 (d, *J* = 8.4 Hz, 1 H)), 7.81 (br s, 1 H), 7.60 (dd, *J* = 8.3, 1.4 Hz, 1 H), 7.57 (br s, 1 H), 7.52 (dd, *J* = 8.2, 1.5 Hz, 1 H), 7.37 (br s, 1 H), 4.25 (t, *J* = 6.2 Hz, 2 H), 3.87 (d, *J* = 6.4 Hz, 2 H), 2.14–2.00 (m, 1 H), 1.79–1.68 (m, 2 H), 1.51–1.40 (m, 2 H), 0.99 (d, *J* = 6.7 Hz, 6 H), 0.93 (t, *J* = 7.4 Hz, 3 H). ^1^^3^C NMR (DMSO-*d*_6_, 68 MHz): *δ* 167.8, 164.1, 151.5, 151.2, 141.8, 140.0, 132.5, 129.7, 125.6, 124.1, 120.3, 120.1, 114.5, 112.0, 75.1, 69.7. 30.9, 28.4, 19.6, 19.1, 14.1. HRMS-ESI (*m*/*z*): [M + Na]^+^ calcd for C_22_H_27_N_3_NaO_6_: 452.1798, found 452.1792.

*4-[(3-sec-Butoxy-4-nitrobenzoyl)amino]-3-isobutoxybenzamide* (**14s**): Light yellow solid, 26 mg, 40% overall yield based on the loading of Fmoc-Rink amide resin, >99% purity by HPLC, m.p. 216–218 °C. ^1^H NMR (DMSO-*d*_6_, 270 MHz): *δ* 9.75 (br s, 1 H), 7.99 (br s, 1 H), 7.97 (d, *J* = 8.4 Hz, 1 H), 7.83 (d, *J* = 8.2 Hz, 1 H), 7.81 (br s, 1 H), 7.59 (d, *J* = 8.2 Hz, 1 H), 7.59 (br s, 1 H), 7.53 (d, *J* = 8.4 Hz, 1 H), 7.37 (br s, 1 H), 4.78–4.71 (m, 1 H), 3.86 (d, *J* = 6.4 Hz, 2 H), 2.11–2.02 (m, 1 H), 1.73–1.62 (m, 2 H), 1.29 (d, *J* = 6.2 Hz, 3 H), 0.98 (d, *J* = 6.7 Hz, 6 H), 0.93 (t, *J* = 7.4 Hz, 3 H). ^1^^3^C NMR (DMSO-*d*_6_, 68 MHz): *δ* 167.8, 164.1, 151.3, 150.6, 142.8, 139.8, 132.5, 129.8, 125.5, 124.3, 120.3, 120.0, 115.3, 112.0, 77.6, 75.1, 29.0, 28.4, 19.6, 19.3, 9.8. HRMS-ESI (*m*/*z*): [M + Na]^+^ calcd for C_22_H_27_N_3_NaO_6_: 452.1798, found 452.1798.

### 4.3. Proliferation Assays

LNCaP cells were obtained from the American Type Culture Collection (ATCC, Manassas, VA) and maintained in T medium (Invitrogen, Carlsbad, CA, USA) supplemented with 5% fetal bovine serum (FBS). All growth media were supplemented with penicillin (100 IU/mL) and streptomycin (100 µg/mL). For androgen deprivation, LNCaP cells were plated (2–10 × 10^3^ per well) in 96-well plates and washed with PBS. The growth medium then changed to phenol-red-free RPMI 1640 with 1%–5% charcoal-stripped fetal bovine serum (CSF) for 48 h. The cells were pretreated with DMSO (vehicle control), or compounds (inhibitors) for 2 h. Media containing ethanol (vehicle control) or DHT was then added to a final concentration of 10 nM and cells cultured for another 72 h. Cell proliferation was measured using the MTT colorimetric assay (Roche Diagnostics, Indianapolis, IN). All experiments were performed in triplicate and the average of experiments displayed.

### 4.4. Western Blot and Immunoprecipitation Analyses

After treatments as indicated, total cellular protein was extracted and Western blotting and/or immunoprecipitation analyses using Protein G Dynabeads (Invitrogen) were performed as previously described [27].

### 4.5. Luciferase Assays

Cells were transfected as indicated with Lipofectamine Plus (Invitrogen), then equally divided into 24-well plates and allowed to attach. The culture medium was replaced after 24 h with androgen deprivation medium containing DHT or vehicle control for 48 h. All experiments were performed in triplicate. Luciferase activity was measured using the Dual Luciferase assay system (Promega, Madison, WI, USA) and normalized to sample protein concentration. Results are presented as fold change over untreated cells. 

### 4.6. Quantitative Real-Time Reverse Transcription Polymerase Chain Reaction (qRT-PCR)

Total cellular RNA was extracted with the RNeasy mini kit according to the manufacturer’s instructions (Qiagen, Valencia, CA, USA). Complementary DNA (cDNA) was then synthesized from 1 µg RNA using the cDNA synthesis kit (Bio-Rad, Hercules, CA, USA). PCR was performed as previously described [10].

### 4.7. Molecular Docking Study

AutoDock Tools 1.5.6 (ADT; The Scripps Research Institute, La Jolla, CA, USA) was used to create input PDBQT files of a receptor and a ligand. The input file of AR was prepared using the published coordinates (PDB 1T63). Water molecules were removed from the protein structure and hydrogens were added. All other atom values were generated automatically by ADT. The docking area was assigned visually around the peptide ligand. The grid box was centered to cover the AF-2 domain of the AR and to accommodate ligand to move freely. The grid box was set to 22 Å × 22 Å × 22 Å, and the x,y,z coordinates of the center of the grid box were set to x = −37.7, y = 25.1, and z = 20.1, respectively. The input files of compound **14d** and **D2** were created from its energy-minimized conformation using ADT. Docking calculations were performed with AutoDock Vina 1.1.2 [22]. A search exhaustiveness of 16 was used and all other parameters were left as default values. The docked conformations were minimized in the MacroModel suite of Maestro (OPLS-2005 force filed, version 9.1, Schrödinger, LLC, New York, NY, USA, 2010). The predicted binding modes were visualized using Maestro.

## 5. Conclusions

In this study, we examined the effects of functional group modifications at the N/C-terminus and the side chains of bis-benzamide **D2**, which were previously found to be effective at inhibiting cell growth of prostate cancer cells (LNCaP). The nitro group at the N-terminus of **D2** appears to be critical for its biological activity. At the C-terminus, a primary carboxamide showed potent growth inhibition as comparable to the methyl ester of **D2**. To survey the optimal substituents at the side chains, we constructed a small bis-benzamide library containing various alkyl groups which differ in substitution pattern and the length of carbon chain. The bis-benzamide library was examined for the antiproliferative activity and identified compound **14d** as the most potent inhibitor with an IC_50_ value of 16 nM. Compound **14d** was found to exert anticancer activity by disrupting the AR–PELP1 interaction and AR transactivation. This study suggests that the bis-benzamide structure is an effective scaffold for producing a chemical library and identifying potent inhibitors of the AR–coactivator interaction.

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
