# Peer review of "A Structure—Activity Relationship Study of Bis-Benzamides as Inhibitors of Androgen Receptor—Coactivator Interaction"

_molecules, 2019, doi:10.3390/molecules24152783_

Round 1

Reviewer 1 Report

The manuscript by Lee et al describes the synthesis and modifications of a battery of novel bis-benzamiudes. Functionally the compounds were tested as peptidomimetics to interfere with coactivator-androgen receptor interaction in the prostate cancer LNCaP cells.

 Authors found bioactivity in the nanomolar range of derivatives with antiproliferative activity. Further, the interaction of the coactivator PELP is strongly reduced by one derivative (14d) that inhibits androgen receptor-mediated transactivation.

The compounds are interesting but they may act not only specific for the androgen receptor. The inhibition of cell proliferation might be due to general toxicity or to other pathways and not necessarily the inhibition of androgen receptor LXLL motif interactions. Since LNCaP cell growth is androgen-dependent it is unclear whether the compounds that reduce cell proliferation are interfering with the androgen –dependent growth.

Major points:

1.     To detect whether there is a general cell toxicity other prostate cancer cells lacking the expression of the androgen receptor must be tested, such as DU145 or PC3 cells using the MTT assay.

2.     To detect whether the androgen-signaling is indeed inhibited, cell growth of LNCaP cells must be tested with and without androgens

3.     To detect whether androgen-mediated transactivation is specifically inhibited, selected compounds shall be analyzed with and without androgen treatment for expression analysis of endogenous androgen receptor target genes, e.g. PSA, or other known target genes.

4.     Authors should explain with compound m14s that the PELP-AR interaction is only slightly weakened while there is a strong inhibition of the luciferase units.

5.     Fig. 2: What is set as 100% if values are 160% for cell viability? Does it suggest growth promoting activity of compound 14d and 14s?

Author Response

Please see the attachment for the point-by-point responses for the comments from the Reviewer #1.

Reviewer 2 Report

I really appreciated the work by Lee and coworkers, not only for what concerns this article, but for the idea that laid the ground to it. Using benzamide derivatives to mimic α-helices is clever due to synthetic feasibility, conformational rigidity and versatility. The present article is a natural prosecution of the ground-braking Tet. Lett. 2007 and Nature Commun. 2013. Although this article is not particularly novel, compared to the other two, it provides additional and useful information (the sentence at page 4, lines 149-150: “These results suggest that the interaction of the bis-benzamides on the AF-2 domain of AR is specific, and topology of the side chain groups is critical for high affinity” is worth the all article).

I only have a few minor corrections to suggest:

1. At page 2, Results section, a figure showing the general formula of the benzamidoaniline moiety evidencing the three area of SAR investigation might be of help

2. At page 8, lines 296-297, is reported: “[M+H]+ calcd for C 25 H 33 N 2 O 6 : 457.23, found

479.79”; why there is this discrepancy?

3. At page 14, section 4.6, the authors state that Autodock 4.2 (AD4) and AutoGrid (AG) software, implemented in ADT, were used to create the pdbqt input and to define the docking area. Formally, this is not correct. AG and AD4 are “stand-alone” packages, that can be integrated into the ADT interface (which is part of the MGLTools package), that compute grid maps (non necessary for vina, that does it on the fly) and perform the docking, respectively. What the authors (probably) did is using ADT to convert the PDB/Mol2 files into pdbqt and to define the docking space through the “Grid” menu. No AG or AD4 calculations were needed for this step, which is only preparatory. As the authors state, the “real” calculation is done in a later step using vina.

Author Response

Please see the attachment for the point-by-point responses for the comments from the Reviewer #2.

Reviewer 3 Report

In this work the authors report the synthesis and study of bis-benzamides as inhibitors of the androgen receptor−coactivator interaction. The introduction is concise, although may benefit of additional referencing. This is an interesting and relevant approach in cancer research and this foldamer-protein interaction is interesting, given the specificity resulting from detailed SAR analysis with a number of bis-benzamides. The experimental section, both in the synthesis, chemical and biological characterization of the compounds is well carried out. Docking calculations require some improvements in reporting methodology and results in terms of number of conformations, binding energy, cluster analysis.

All in all, although this is a continuation of previous research about the discovery of D2, that is herein reported as a reference compound, we think this paper merits publication to Molecules following some revisions:

Scheme 1 should report reaction yields at every step, and so as for overall yield of scheme 2.

Figure 1 may report the general structure of bis-benzamides with R1 and R2, to better read the histogram reporting cell viability results.

In Autodock calculation the grid seems rather small as compared to the size of the molecule. Please comment this.

Also, as the interactions of both 14d and D2 seem rather solvent-exposed, the results may not be fully reliable. What about sampling in docking calculation? How many conformers were generated? What about the mean binding energy and cluster analysis?

Also, the minimum energy docked conformation should be minimized with the receptor to achieve full convergence of the calculation.

All this part require revision to allow for discussing reliable data resulting from docking calculations.

Author Response

Please see the attachment for the point-by-point responses for the comments from the Reviewer #3.

Round 2

Reviewer 1 Report

Authors have addressed satisfactory to full level the raised critical points.